# A Chinese BERT-Based Dual-Channel Named Entity Recognition Method for Solid Rocket Engines

**Zhiqiang Zheng, Minghao Liu and Zhi Weng ***

College of Electronic and Information Engineering, Inner Mongolia University, Hohhot 010021, China
* Correspondence: wzhi@imu.edu.cn

**Abstract:** With the Chinese data for solid rocket engines, traditional named entity recognition cannot be used to learn both character features and contextual sequence-related information from the input text, and there is a lack of research on the advantages of dual-channel networks. To address this problem, this paper proposes a BERT-based dual-channel named entity recognition model for solid rocket engines. This model uses a BERT pre-trained language model to encode individual characters, obtaining a vector representation corresponding to each character. The dual-channel network consists of a CNN and BiLSTM, using the convolutional layer for feature extraction and the BiLSTM layer to extract sequential and sequence-related information from the text. The experimental results showed that the model proposed in this paper achieved good results in the named entity recognition task using the solid rocket engine dataset. The accuracy, recall and F1-score were 85.40%, 87.70% and 86.53%, respectively, which were all higher than the results of the comparison models.

**Keywords:** solid rocket engines; named entity recognition; BERT pre-trained language model; dual-channel network model

## 1. Introduction

Conducting research on solid rocket engines (SREs) can provide more power options for launching vehicles and is of great significance in promoting the development of launch vehicle technology. Launch vehicle technology is not only the preliminary basis for space activities but also an important manifestation of national defense strength. The Chinese SRE literature and the related scientific research experiments have generated a large amount of data, which are stored in the form of unstructured text. If we can effectively use these text data, extracting effective information from unstructured data to transform them into structured data, they can be employed to serve researchers and industry enthusiasts. The information extraction techniques that have been applied to meet this need have aimed to automate the extraction of information using computer technology [1]. Currently, named entity recognition (NER) is, as an information extraction task, gradually developing into a more mature and complete method. In the field of SREs, this means the accurate extraction of relevant entities from textual data; for example, terms such as "rocket engine" and "solid propellant" can be identified as solid rocket motor-related entities. Furthermore, concepts such as carbon dioxide, cyanide and ammonium nitrate can be recognized as chemical entities.

NER was first proposed in the sixth Message Understanding Conference (MUC-6), in which the naming entities studied mainly included the names of people, locations and organizations [2]. NER is one of several natural language processing (NLP) tasks, and it is a key technology in such tasks, which include information retrieval, automatic text summarization, intelligent questions and answers, machine translation and knowledge base construction [3].

The NER task can be effectively implemented using deep learning techniques. However, deep learning is a representation learning algorithm based on large-scale data and

machine learning, and acquiring a large amount of data is very time-consuming and labor-intensive. Transfer learning can largely resolve the issue of deep learning requiring large amounts of data to support it [4]. Transfer learning makes it possible to learn new tasks while incorporating previously learned knowledge. It achieves this by drawing on the process of human learning in a way that builds on previous learning to achieve better and faster results using only a small amount of data.

Due to the extensive information diversification in society, domain-specific NER has become more valuable for practical applications. For example, research in the medical, agricultural and electric power fields has gradually expanded to involve more than a dozen languages, such as Chinese, Japanese and Hindi. Currently, its main technical difficulty is that the entity boundary in text data is ambiguous. In the open literature on SREs, there is no mention of an entity recognition corpus or a unified annotation strategy. Therefore, it is especially important to develop corpus annotation rules. In this study, a high-quality SRE Chinese corpus was constructed based on expert experience. To obtain a tightly connected proprietary vocabulary, we used the BERT pre-trained language model to extract contextual text features from the input Chinese corpus, and we then applied them jointly with CNN and LSTM models for our NER technology research. The specific contributions are as follows:

- High-quality SRE datasets were constructed by acquiring knowledge on solid rocket engines from open Internet resources using crawling techniques and by combining expert experience and rules;
- For the first application of a BERT pre-trained language model to the SRE-NER task, we developed a dual-channel BERT-based network architecture;
- Different models were used for the SRE-NER task, and the efficiency of the models was compared.

The remainder of this paper is structured as follows. Related work is presented in Section 2. Section 3 introduces the current difficulties and problems in this area, describes the construction of the dataset and provides specific details about the BERT-based dual-channel network architecture. Sections 4 and 5 detail the evaluation criteria and the experiments conducted. Finally, Section 6 discusses the conclusions drawn from the experiments and our focus for future research in this area.

## 2. Related Work

### 2.1. Named Entity Recognition

2.1.1. Traditional NER Methods

The traditional methods for NER are currently divided into rule-based and unsupervised learning approaches.

Rule-based NER systems rely on human-developed rules. The rules can be self-designed according to the required research area. In 2020, Raabia Mumtaz et al. proposed an NER system called CustNER that combines the existing NER and DBpedia knowledge bases for person, location and organizational entity recognition [5]. In the same year, Pushpalatha et al. proposed a rule-based method for the Kannada textual body NER method that divides sentences into different words using data preprocessing to identify Kannada named entities [6].

One typical method that is used for unsupervised learning is called clustering. A cluster-based NER system extracts named entities from clustered groups based on contextual similarity. In 2013, Shaodian Zhang and Noémie Elhadad proposed an unsupervised method for extracting named entities from biomedical texts. Their model uses terminology, corpus statistics and shallow grammatical knowledge. The effectiveness and generalizability of their method were demonstrated with two biomedical datasets [7]. In 2022, Senthamizh Selvan proposed an entity-aware text summarization technique based on document clustering that can extract summaries from multiple documents. The extracted entities were then sorted according to Zipf's law and clustered using K-means to form sentence clusters [8].

### 2.1.2. Deep Learning-Based NER Methods

Unlike feature-based approaches, deep learning network models can spontaneously mine data for hidden features. The BiLSTM-CNN model, which was proposed by Chiu and Nichols, combines a bidirectional LSTM and a character-level CNN. In addition to word embedding, the model uses additional word-level and character-level features [9]. Shanta Chowdhury et al. proposed a multi-task bidirectional RNN model for Chinese electronic medical record entity extraction. They experimentally demonstrated that the model improved the micro average F-score, the macro average F-score and the overall accuracy by 2.41%, 4.16% and 5.66%, respectively, in comparison to the benchmark model [10]. In 2021, Goyal Archana et al. proposed a new NER system that combines enhanced word embedding and a deep learning approach. Enhanced word embeddings (EWEs) were generated by cascading FastText word embeddings with minimal feature embeddings, and they performed well in Hindi, Punjabi and bilingual Hindi NER [11]. In 2020, Hong et al. proposed a novel CRF-based framework called the DTran NER. The DTran NER uses two independent deep learning-based networks: the unary network and the pairwise network. The results showed that the introduction of the deep learning-based label transformation model provided contextual cues for enhancing the Bio-NER that were different from those provided by the static transformation model. This model also performed well with public datasets [12].

### 2.2. Transfer Learning

Transfer learning can be leveraged to improve NER accuracy by transferring the learned knowledge to the relevant problems. In 2017, Google proposed the Transformer, the structure of which is shown in Figure 1. It can achieve better performance without using a sequence-aligned cyclic architecture. Instead, it simply uses self-attention and a feed-forward neural network. The performance and the Transformer can be generalized to other tasks [13].

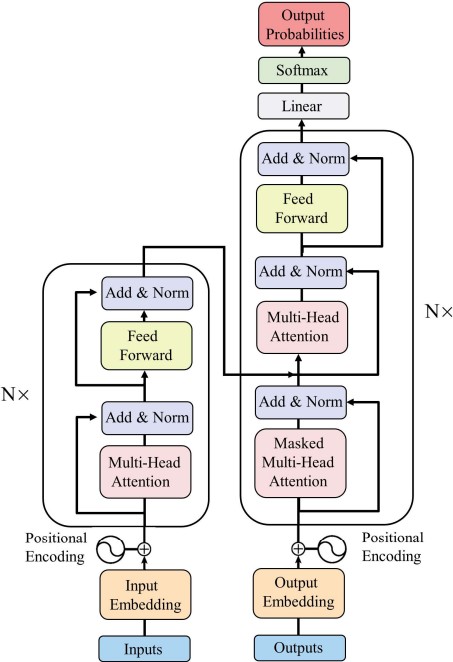

**Figure 1.** The structure of the Transformer [13].

The ELMo model was proposed in March 2018. The model infers the word vectors corresponding to each word based on the context and is able to understand the meanings of multiple words based on the context. It significantly outperformed the SOTA with regard to six NLP problems [14]. The Google AI language team proposed the BERT pre-trained

language model, and it performed well in SQuAD1.1: it outperformed humans across the board in two measures and represents the SOTA for 11 different NLP tests [15]. CMU and Google Brain collaborated to introduce XLNet, which was found to significantly improve the performance of tasks in the reading comprehension category for long texts and generated better results in industrial scenarios and machine translation [16]. In 2022, Ankit Agrawal et al. proposed a solution to the entity nesting problem that arises in NER using a transfer learning approach. In this study, they solved the problem by using a joint-labeling modeling technique combined with fine-tuning and pre-training of BERT. The transfer learning-based approach obtained better results for the nested NER task throughout the study [17].

### 2.3. Artificial Intelligence and Solid Rocket Engines

In recent years, there has been an increasing amount of research combining artificial intelligence with the study of rocket engines. This is an important contribution to the development of the rocket engine field. In 2020, Aimee Williams et al. combined input measurement data and machine learning to create a "virtual sensor" that can provide critical information that traditional measurement methods cannot obtain due to the inability to place sensors in the combustion chamber [18]. In 2021, Dongxu Liu et al. proposed and investigated a deep convolutional neural network (CNN) architecture that uses the finite element method (FEM) to generate labeled training data to evaluate the scale of defects in solid rocket engines coexisting with bore cracks and propellant debonding [19]. In the same year, Dhruv Gamdha et al. used CNN techniques for the automatic detection of anomalies in X-ray images of solid propellant pillars. This simulation system is capable of both defect detection and defect instance segmentation. The simulation results showed an accuracy of more than 87% when it was applied to a test set that contained 416 images [20]. In 2022, Surina proposed an imaging-based deep learning tool for measuring fuel regression rates in two-dimensional plate burner experiments with mixed rocket fuels. The network was superior to conventional image processing techniques in filtering soot, pitting and wax deposition on the chamber glass and in filtering flame-introduced noise [21]. In this study, we applied natural language processing techniques to the field of solid rocket motors. This study provides an efficient information extraction framework for entity-specific word extraction.

### 2.4. Domain-Specific Named Entity Recognition

In recent years, deep learning has greatly improved the performance of NER models. Due to practical needs, more scholars have shifted their attention to domain-specific tasks. In 2021, Lei Yang et al. proposed a BIBC-based NER method that focuses on the recognition of diabetic entities. The models used in this study were mainly the BERT-WWM and IDCNN-BiLSTM-CRF modules. The method could be used to extract diabetic entities accurately and met the requirements of practical applications [22]. In 2022, Yuqing Yu et al. proposed a mineral NER model based on deep learning to construct a knowledge map in the mineral domain. The BERT-CRF algorithm was used, and the final experimental results showed that the model could effectively identify seven mineral entities with an average F1-score of 84.2% [23]. In the same year, Guo Xuchao et al. proposed a new model based on enhanced contextual embedding and glyph features for agricultural NER. The experimental results showed that the model obtained 95.02% and 96.51% F1-scores with the AgCNER and Resume datasets [24], respectively. Sun Junlin et al. constructed a training and evaluation model for a natural disaster-annotated corpus and proposed a natural disaster NER method based on the XLNet-BiLSTM-CRF model that obtained 92.80% precision, 91.74% recall and a 92.27% F1-score [25]. Min Wang et al. proposed a multi-feature-based character-level entity recognition model for power texts. The experimental results showed that the total F1-score of the model was improved by 2.26% and that the recognition accuracy of each label was also improved [26]. An Fang et al. used different models

for the extraction task with tumor entities and found that the BERT-BiLSTM-CRF model was superior to the various other models, achieving 95.57% accuracy [27].

The above researchers applied pre-trained models well in domain-specific NER tasks; however, all used single-branch networks in the subsequent network structures. In this study, NER was applied to the Chinese SRE domain, and it not only extracted character features effectively using pre-trained language models but also combined the advantages of two-channel network models, fusing the two channels for decision-making purposes. This made the performance of our model for Chinese SRE data superior to that of other models.

## 3. Methodology

The difficulties encountered in this experiment in the process of SRE corpus construction and the development of the NER network model are introduced in detail in this section. Section 3.1 highlights the current difficulties in constructing datasets in the SRE domain. Section 3.2 proposes a method to solve the problem of constructing SRE domain-specific datasets. Section 3.3 introduces the architecture of the model used in this study.

### 3.1. Problem Formulation

In the field of SREs, entity classification is complex and entity annotation rules are highly specialized and complex. In addition, there are a large number of long words, which leads to entity nesting problems. Having a good database is the key to conducting experiments, and solving the above problems effectively is the key to achieving accurate NER.

### 3.2. Dataset Construction

In this study, we obtained publicly available data related to solid rocket engines from the Web through Web crawler techniques. Three representative books were used: *Solid Rocket Engine Design*, *Solid Rocket Engine Principles* and *Rocket Engine Fundamentals*. Before constructing the dataset, domain experts constructed annotation rules according to the specificity of the data. Since this dataset was intended to focus on SRE-related professional entities, the dataset labels were divided into seven categories. The BIO dataset construction method commonly used in the named entity identification domain was stringently adopted to construct the solid rocket motor domain dataset. The dataset was constructed on an exact factual basis and no structural properties were assumed for the dataset. This is shown in Table 1.

**Table 1.** Solid rocket engine data labeling.

| Marking Symbols | Meaning |
| --- | --- |
| B-SRE | The beginning of the solid rocket engine entity |
| I-SRE | The interior of the solid rocket engine entity |
| B-CHEMPHY | The beginning of physical chemistry nouns |
| I-CHEMPHY | The interior of physical chemistry nouns |
| B-PER | The beginning of the person entity |
| I-PER | The interior of the human-named entity |
| O | Non-entity words—not related to the above |

However, since there are no uniform annotation rules for entity identification data in this domain, different annotators have different annotation situations when annotating, as shown in Table 2.

In this study, we constructed a corpus by combining experts' experience. Multiple annotators annotated the same data. The data with the same annotation results were stored in the corpus. The annotated data for which there were inconsistent opinions were corrected by domain experts, and these modified data were then stored in the corpus. By combining the opinions of domain experts, the problems of entity classification and boundary ambiguity in the SRE domain could be solved more effectively. As Figure 2 shows, the corpus annotation and corpus quality enhancement methods were based on experts' experience.

**Table 2.** Examples of ambiguity in markup results.

| Marker 1 | Marker 2 | Marker 3 |
|----------|----------|----------|
| 燃 * B-SRE | 燃 B-SRE | 燃 B-CHEMPHY |
| 烧 I-SRE | 烧 I-SRE | 烧 I-CHEMPHY |
| 室 I-SRE | 室 I-SRE | 室 O |
| 燃 B-CHEMPHY | 燃 B-CHEMPHY | 燃 B-CHEMPHY |
| 气 I-CHEMPHY | 气 I-CHEMPHY | 气 I-CHEMPHY |
| 压 I-CHEMPHY | 压 B-CHEMPHY | 压 I-CHEMPHY |
| 强 I-CHEMPHY | 强 I-CHEMPHY | 强 I-CHEMPHY |
| 达 O | 达 O | 达 O |
| 到 O | 到 O | 到 O |
| 平 B-CHEMPHY | 平 O | 平 B-CHEMPHY |
| 衡 I-CHEMPHY | 衡 O | 衡 I-CHEMPHY |
| 压 B-CHEMPHY | 压 B-CHEMPHY | 压 I-CHEMPHY |
| 强 I-CHEMPHY | 强 I-CHEMPHY | 强 I-CHEMPHY |

* Part of the Chinese data set about solid rocket engines.

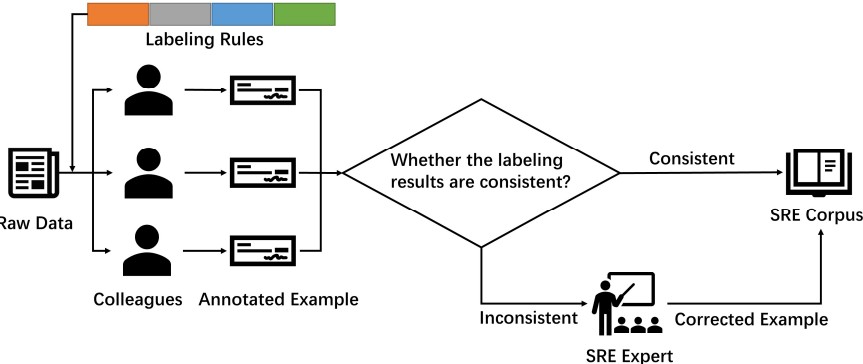

**Figure 2.** Arbitration-based approach to corpus annotation and corpus quality enhancement.

### 3.3. Architecture of the Proposed System

The NER model proposed in this paper mainly consists of a BERT word embedding module, a dual-channel module and a CRF module, and its overall architecture is shown in Figure 3. The BERT pre-trained language model was first used to encode individual characters, through which a vector representation corresponding to each character was obtained. Next, a dual-channel model was designed to semantically encode the input text. Finally, the labels with the highest probability were output through the CRF layer to obtain each character's class. The details of each module are described below.

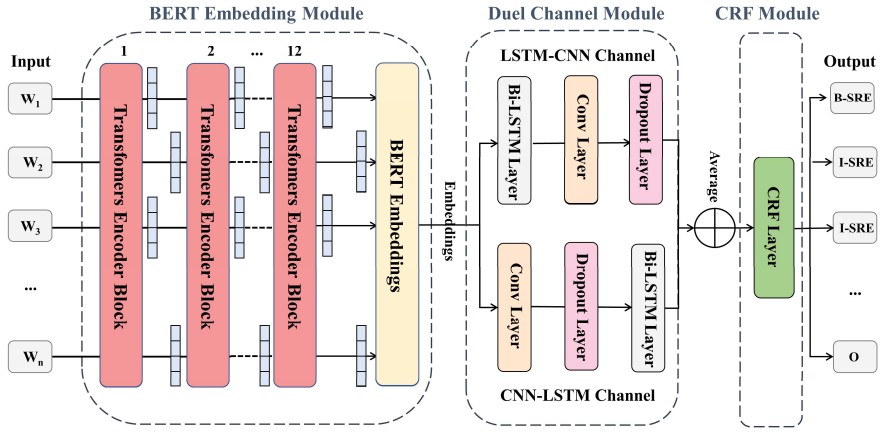

**Figure 3.** BERT-based dual-channel named entity recognition model.

### 3.3.1. BERT Embedding Module

First, the labeled text data for the SRE domain with tags D = {d1, d2, d3, . . . , dn } and L = {l1, l2, l3, . . . , ln } were input into the word embedding module, which contained the BERT pre-trained model. The pre-training parameter profile for BERT was the Chinese version of BERT published by Google: "Chinese_L-12_H-768_A-12". The model structure of the BERT-based model was a multi-layer bidirectional Transformer structure. In the encoder part, the BERT-based model had 12 attention heads, 768 hidden units and an encoder module that contained 12 Transformers. In the attention mechanism, each character corresponded to three different vectors: a query vector (Q), key vector (K) and value vector (V). These three vectors were obtained by multiplying the embedding vectors by three different weight matrices: wq, wk and wv. The query vector was then multiplied by the key vector score of each word. Note that the value was the score term obtained using the softmax function and the result was multiplied by the value vector [28] as follows:

$$Attention(Q, K, V) = softmax\left(\frac{QK^T}{\sqrt{d_k}}\right) \tag{1}$$

In addition, the Transformer coding unit added residual networks and layer normalization to address the degradation problem as follows:

$$LN(x_i) = \alpha * \frac{x_i - \mu_L}{\sqrt{\sigma_L^2 + \epsilon}} + \beta \tag{2}$$

$$FNN = \max(0, xW_1 + b_1)W_2 + b_2 \tag{3}$$

where $\alpha$ and $\beta$ are the parameters to be learned, and $\mu$ and $\sigma$ are the mean and variance of the input. The embedding function converted the input characters to their respective embedding. The intermediate encoded representation included the position, segment and token of the embedded input word and passed it to the next encoder block. The intermediate encoded representations were combined in the final BERT embedding.

### 3.3.2. Dual-Channel Module

After the embedding module generated the encoding—E = {e1, e2, e3, . . . , en}—for each character, including for the location and the contextual information, the encoding was fed into the dual-channel module. This dual-channel module was a combination of two channels: LSTM-CNN and CNN-LSTM.

The CNN-LSTM module extracted the text features well and provided them to the BiLSTM layer, which finally output the individual label probability values of the characters from this branch network. In the CNN-LSTM model, the input of the CNN was the embedding of characters generated by BERT. The dimensions of this embedding were [N, 768]. The output was a vector of [N, 369, 32]. The input of the BiLSTM was the output of the CNN layer after dropout. Its output was the score value of the predicted label for each character calculated using softmax. For example, the outputs of the character $W_1$ at the BiLSTM node were 1.5 (B-SRE), 0.9 (I-SRE), 0.1 (B-CHEMPHY), 0.08 (I-CHEMPHY) and 0.05 (O).

The LSTM-CNN module passed the embedding generated by BERT through the LSTM layer first and then fed it to the CNN network, which simultaneously generated label prediction values that were different from the other channel. The input of the BiLSTM was the embedding of the BERT and the output was a vector of [N, 768, 32]. This vector was sent through the CNN layer next. The output of the CNN was the fractional value of the predictive labeling obtained using softmax for each character. For example, the outputs of the character $W_1$ at the CNN node were 1.0 (B-SRE), 1.1 (I-SRE), 0.3 (B-CHEMPHY), 0.07 (I-CHEMPHY) and 0.06 (O).

By combining and averaging the outputs of the two channels, the advantages of the two branching networks were effectively utilized to form the output results of the dual-

channel module. Then, the output decisions were taken by the CRF module to achieve the final classification task, including a dropout layer with a dropout of 0.5 after the convolutional layer to prevent overfitting. The idea behind using the dual-channel model was to take advantage of the convolutional layer for feature extraction while employing the BiLSTM layer to extract text order and sequence-related information. Using the combined dual-channel module helped the system to efficiently learn both features and to sequence information from the input text.

### 3.3.3. CRF Module

The CRF module is a verdict-based model that adds constraints to final prediction labels to ensure validity. During training, the layer can automatically learn these constraints from the training dataset. After obtaining the hidden layer vector from the output of the two-channel module, the label with the highest probability is output to obtain the class of each character. In this study, for the specified sequence D = {d1, d2, d3, . . . , dn } and its corresponding labels L = {l1, l2, l3, . . . , ln }, the score was defined as follows:

$$S(X, y) = \sum_{i=0}^{n} A_{y_i, y_{i+1}} + \sum_{i=1}^{n} P_{i, y_i} \tag{4}$$

where $A$ is the transfer score matrix and $A_{i,j}$ denotes the score transferred from label $i$ to label $j$. The maximum probability of the sequence label $y$ was calculated using the softmax function as follows:

$$P(y|X) = \frac{e^{S(X,y)}}{\sum_{\widetilde{y} \in Y_X} e^{S(X,\widetilde{Y})}} \tag{5}$$

$$\log(P(y^*|X)) = S(X, y^*) - \log\left(\sum_{\widetilde{y} \in Y_X} e^{S(X,y)}\right) \tag{6}$$

where $Y_X$ is the sequence of all possible labels for the input sentence X. In the decoding process, the Viterbi algorithm was used to decode and, finally, the sequence with the highest total predicted score was output as the final optimal sequence as follows:

$$y^* = argmax_{\widetilde{y} \in Y_X}(S, \widetilde{y}) \tag{7}$$

Before training the model, we randomly initialized the scores of the transition matrix. These scores were updated through the iterative process of training. Among other things, the transition matrix of the CRF was trained by a loss that contained global information and broke first-order Markovianity. To make the transition matrix more robust, we added two types of labels: START for the beginning of a sentence (not the first word of a sentence) and END for the end of a sentence. The following Table 3 shows the transition score matrix with the START and END labels.

**Table 3.** Transition matrix.

|  | START | B-SRE | I-SRE | B-CHEMPHY | I-CHEMPHY | B-PER | I-PER | O | END |
|---|---|---|---|---|---|---|---|---|---|
| START | 0 | 0.8 | 0.03 | 0.7 | 0.0009 | 0.8 | 0.007 | 0.9 | 0.08 |
| B-SRE | 0 | 0.6 | 0.9 | 0.2 | 0.0006 | 0.3 | 0.0002 | 0.7 | 0.005 |
| I-SRE | −1 | 0.5 | 0.53 | 0.55 | 0.002 | 0.48 | 0.0003 | 0.86 | 0.004 |
| B-CHEMPHY | 0.9 | 0.0002 | 0.06 | 0.3 | 0.8 | 0.07 | 0.008 | 0.43 | 0.006 |
| I-CHEMPHY | −0.9 | 0.2 | 0.008 | 0.7 | 0.09 | 0.08 | 0.0005 | 0.77 | 0.2 |
| B-PER | 0 | 0.03 | 0.01 | 0.0004 | 0.07 | 0.5 | 0.9 | 0.6 | 0.009 |
| I-PER | −0.3 | 0.5 | 0.02 | 0.008 | 0.01 | 0.7 | 0.09 | 0.8 | 0.08 |
| O | 0 | 0.45 | 0.003 | 0.7 | 0.0008 | 0.65 | 0.0007 | 0.9 | 0.004 |
| END | 0 | 0 | 0 | 0 | 0 | 0 | 0 | 0 | 0 |

As shown in the table above, the transfer matrix learned some useful constraints:

- The first word of the sentence should be "B-" or "O" and not "I" because the transfer score from "START" to "I-SRE or I-CHEMPHY" was very low;

- "B-label 1, I-label 2, I-label 3 . . . ", in which the categories 1, 2, 3 should be the same entity category. For example, "B-SRE, I-SRE" was correct while "B-SRE, I-CHEMPHY" was wrong because the transfer score from "B-SRE" to "I-CHEMPHY" was very low;
- "O I-label" was wrong, as the named entity should start with "B-" instead of "I-".

In our proposed NER model, the input sentences were first pre-trained with BERT to generate a corresponding pre-trained word-embedding from the sentences. Words were then embedded into the input dual-channel network to further extract character features and contextual information. The model effectiveness could be significantly improved using the advantages of the dual-channel network model. Finally, the probability value of the dual-channel output was plugged into the CRF layer containing the sequence transfer probabilities, and constraints were added to the final prediction labels to ensure the correctness of the final annotation results. The input sentence was processed in the model as described in Figure 4.

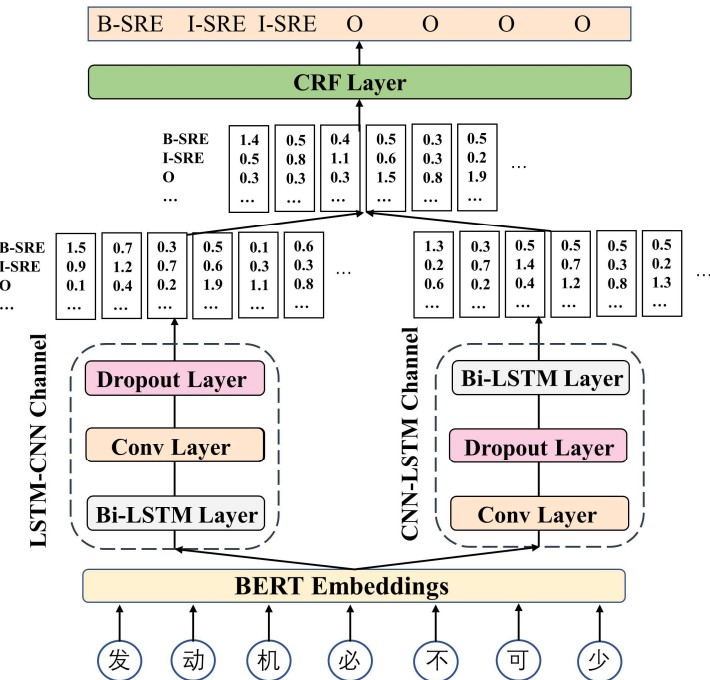

**Figure 4.** Input example for the model (The input was Chinese data).

## 4. Experiments

### 4.1. Dataset

Unlike traditional natural language processing tasks, which use public corpora, the SRE domain dataset that was used in this study does not have a high-quality dataset. We constructed the SRE dataset using the method described in Section 3.2. It consisted of various related literature data (approximately 170,000 words). The self-built experimental corpus was processed at the sub-sentence level to obtain 6287 sentences, and the dataset was divided into training, testing and validation sets in the ratio 8:1:1. The statistics for the numbers of entities in each category are shown in Table 4.

**Table 4.** Named entity dataset.

| Entity Category | Training Set | Test Set | Validation Set | Number of Entities |
|---|---|---|---|---|
| SRE | 1156 | 204 | 169 | 1529 |
| Physical and Chemistry | 1270 | 187 | 155 | 1612 |
| Person | 1087 | 166 | 181 | 1434 |

### 4.2. Evaluation Metrics

To validate the model, its predicted output needed to be compared with the results of the true labeling to measure the model's performance. The process of predicting output labels from the input sequence of an NER task is a typical multiclassification problem in machine learning. The confusion matrix is a common evaluation metric in multiclassification tasks. In the SRE dataset we constructed, the accuracy often did not reflect the advantages of the model because the number of non-entity characters was larger than the number of entity characters. We had to advance further through the confusion matrix to obtain the precision, recall and F1-score, as shown in Table 5. These three metrics could reflect the accuracy of the model in the study.

**Table 5.** Classification results.

| Confusion Matrix | | Predictive Value | |
|---|---|---|---|
| | | Positive | Negative |
| Actual Value | True | TP | FN |
| | False | FP | TN |

1. True positive (TP)—the true result of the label is positive and the predicted result of the model is positive;
2. False positive (FP)—the true result of the label is negative and the predicted result of the model is positive;
3. False negative (FN)—the true result of the label is positive and the predicted result of the model is negative;
4. True negative (TN)—the true result of the label is negative and the model predicts a negative result.

From these four metrics, standard evaluation metrics—precision ($P$), recall ($R$) and F1-score ($F1$)—were used to evaluate the NER model.

$$Precision(P) = \frac{TP}{TP + FP} * 100\% \tag{8}$$

$$Recall(R) = \frac{TP}{TP + FN} * 100\% \tag{9}$$

$$F1 - Score = 2 * \frac{P * R}{P + R} * 100\% \tag{10}$$

### 4.3. Experimental Environment

In this study, to ensure the smooth running of the whole experiment, the environment configuration shown in Table 6 was used.

**Table 6.** Experimental environment.

| Category | Configuration |
|---|---|
| Hardware | CPU: Intel®Core(TM) i9-9900K<br>GPU: NVIDIA Quadro P6000 |
| Software | CUDA: 10.1<br>Python: 3.6<br>TensorFlow: 1.14<br>Numpy: 1.19.2 |

### 4.4. Parameter Setting

The models used in this study were all built using TensorFlow. The default setting for the BERT pre-trained language model uses a 12 head attention-mechanism Transformer

with 768 hidden-layer dimensions and a total of 110 M parameters. The maximum sequence length used in this study was 512, the batchsize was 32, there were 32 LSTM hidden units and 32 CNN filters and the size of each filter was 400. Adam was used as the optimizer, the dropout was set to 0.5 and the learning rate was set to $10^{-5}$.

## 5. Experimental Results and Analysis

In order to assess the performance of the model proposed in this paper in the SRE domain NER task, the effectiveness of the proposed model was compared with the baseline model through comparative experiments. The proposed model was also compared with other models that have previously implemented the NER task in specific domains. In addition, all the compared models in this study (baseline models) were trained with the same dataset (constructed in this study) and experimental setting.

### 5.1. Comparison with Baseline Models

In this study, two baseline models were used for the comparison experiments: the BERT-BiLSTM-CRF model and the BERT-CNN-BiLSTM-CRF model. The BERT-BiLSTM-CRF model became the most widely used model after the BERT was proposed. Based on this baseline model with the addition of CNN layers, the BERT-CNN-BiLSTM-CRF model is generally considered to be the second most widely used baseline model. The BERT-based dual-channel model proposed in this study was compared with the two main baseline models.

As shown in Figure 5, the F1-score for the model proposed in this paper was 2.19% higher than the score for the BERT-BiLSTM-CRF model. It was 1.29% higher than that for the BERT-CNN-BiLSTM-CRF model.

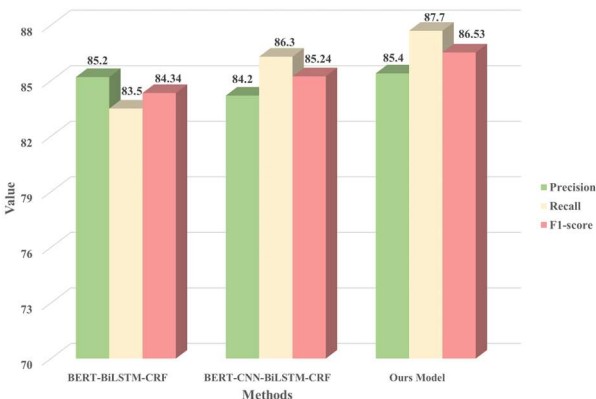

**Figure 5.** Performance of the model in this study compared with the two baseline models.

### 5.2. Comparison of LSTM Unit Dimensions

The dimensionality of the hidden units in LSTM affects the overall performance and computational complexity of the model. In order to obtain the best hyperparameters, hidden units of different dimensions were set. As can be seen from Table 7, our model had the highest F1-score at 86.5% when the dimensions of its hidden cells were 32. However, its F1-score was 1.5% lower than those of the models that had hidden cell dimensions of 32 when its hidden cell dimensions were lower than 32. Furthermore, when the dimensionality of its hidden cells was greater than 32, the F1-score of the model was 2.1% lower than those of the models that had hidden cell dimensions of 32. It was concluded that too few hidden units in our model would lead to an insufficient feature capture capability and, thus, the overall performance of the model would be poor. On the other hand, an increase in the training parameters with increasing dimensionality led to an increase in the computational complexity but poorer performance. Therefore, the LSTM hidden unit size was set to 32.

**Table 7.** Performance comparison with different LSTM cell sizes.

| Dimensions | Precision | Recall | F1-Score |
|:---:|:---:|:---:|:---:|
| 16 | 85.2 | 84.8 | 85.0 |
| 32 | 85.4 | 87.7 | 86.5 |
| 64 | 83.7 | 85.5 | 84.6 |
| 128 | 83.3 | 85.1 | 84.2 |

*5.3. Comparison of Using Dropout*

In addition, a set of comparison experiments were used to verify the effect of the dropout layer. One of the experiments included a dropout layer and the other did not. The results are shown in Table 8. The experiment with the dropout layer exhibited a 2.1% performance improvement compared to the one without.

**Table 8.** Experimental results with and without dropout.

| Dropout | Precision | Recall | F1-Score |
|:---:|:---:|:---:|:---:|
| Not using | 85.6 | 83.2 | 84.4 |
| Using | 85.4 | 87.7 | 86.5 |

*5.4. Performance Comparison with Different Numbers of Epochs*

In order to obtain better model parameters, different numbers of epochs were set: 50, 100, 150, 200 and 300. Table 9 provides a comparison of the model performance with different periods. From the table, it can be seen that, with 150 epochs of training, the model obtained the best values for both the accuracy and F1-score evaluation metrics at 85.4% and 86.5%, respectively. The recall rate was the highest when the number of epochs equaled 200. Therefore, 150 epochs was chosen for the final training of the model.

**Table 9.** Experimental results with different epochs.

| Epoch | Precision | Recall | F1-Score |
|:---:|:---:|:---:|:---:|
| 50 | 84.3 | 85.1 | 84.7 |
| 100 | 84.8 | 87.4 | 86.0 |
| 150 | 85.4 | 87.7 | 86.5 |
| 200 | 85.0 | 86.3 | 85.6 |
| 300 | 84.7 | 85.8 | 85.2 |

*5.5. Comparison between Different Methods*

A set of comparison experiments were also conducted in this study. The models involved in the comparison were the BiLSTM-CRF model and the improved BiLSTM-Attention-CRF model. The attention mechanism can assign different weight coefficients to the vectors of different features in the text to better extract features and, thus, improve the NER performance.

The experiments involving the two models used the self-built SRE dataset. The experimental results are shown in Figure 6. It is obvious that the F1-score for the network model proposed in this paper was 5.29% higher than the BiLSTM-CRF model and 3.74% higher than the BiLSTM-Attention-CRF model. Overall, the model used in this paper achieved the best results in terms of the evaluation metrics. The experiments proved that the model proposed in this paper is efficient in the SRE domain NER task.

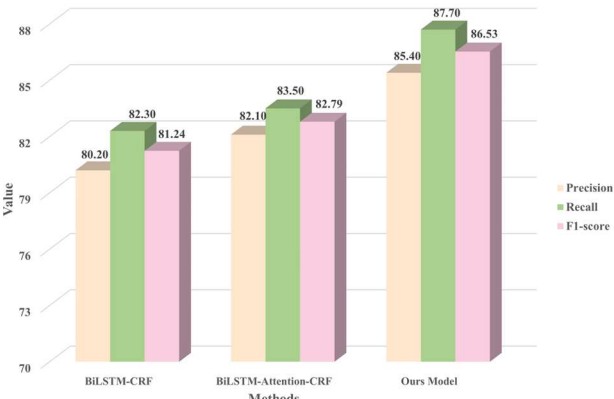

**Figure 6.** Experimental results for the proposed model and the baseline model.

## 6. Conclusions

In this paper, we proposed a Chinese BERT-based dual-channel NER algorithm that can be used in the field of solid rocket engines. The model includes a BERT word embedding module, a dual-channel module and a CRF module. The experimental results from this study showed that the framework proposed in this paper can effectively mine the entity nouns in the SRE domain. Compared with the traditional algorithm, the accuracy, recall and F1-score of our model were significantly higher. With regard to its practical applications, our model also achieved a high recognition rate. From experimenting with publicly available datasets, we believe that the approach described in this paper is applicable in other domains.

However, the method proposed in this paper was based on the use of a small-scale dataset for testing. Therefore, the model has some limitations. In our future work with this model, the dataset size will be expanded. In addition, experiments on language models other than BERT, such as ELMO and ALBERT, will also be conducted in our future work. Since our proposed model has a complex network structure, consuming a large amount of computational resources, making the model lightweight will be another important consideration in later studies.

**Author Contributions:** Conceptualization, M.L. and Z.Z.; methodology, M.L. and Z.W.; software, Z.Z.; validation, Z.Z. and Z.W.; formal analysis, Z.Z.; investigation, M.L.; resources, Z.W.; data curation, M.L.; writing—original draft preparation, Z.Z.; writing—review and editing, Z.Z. and Z.W.; visualization, Z.Z.; supervision, Z.W.; project administration, Z.Z.; funding acquisition, Z.W. All authors have read and agreed to the published version of the manuscript.

**Funding:** This research funding was obtained from the Key Technology Research Plan Project of Inner Mongolia Autonomous Region under grant numbers 2020GG0185 and 2021GG0160.

**Data Availability Statement:** Not applicable.

**Conflicts of Interest:** The authors declare no conflict of interest.

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
