# Peer review of "A Chinese BERT-Based Dual-Channel Named Entity Recognition Method for Solid Rocket Engines"

_electronics, doi:10.3390/electronics12030752_

Round 1

Reviewer 1 Report

The paper proposes a Bert Based approach for NER in Solid Rocket Engines. However, this paper needs a major revision, before assessment:

-Explain the CNN layer, size of inputs and outputs, how the embedding of each character is computed. Do you compute a unique vector for each character or a global vector for the whole text

-Explain the BILSTM layer in terms of input and outputs

-How do you merge the embedding of LSTM-CNN and CNN-LSTM

-Explain the parameters of the CRF layer and their size. Show the final transition matrix

-Do you use fine tuning or pre training of BERT

-How is the model is trained.

-Remove all the repetition, e.g, NER instead of Named entity recognition and BERT instead of .... , BERT is explained in many places

-A lots of typos shoud be fixed such as a space after '.', a sentence start with a capital letter

-

Reviewer 2 Report

The work presented in the paper is interesting in its approach, which proposes a model that leverages BERT for NER in the SRE and proposes a small dataset.

-       The paper needs to be extensively reviewed regarding its English. For example:

o  Do not use the long name again when you define terminology and its acronym. Stick to the abbreviation.

o   Be consistent, and leave a space between a reference number and a sentence. Similarly, leave a space between the first letter of a new sentence and the period ending the previous sentence.

o   Always start the sentence with a capital letter!

o   Only capitalise what needs to be capitalised (e.g. names). Words such as network not within a name or title do not need to be capitalised.  

o   Improve the way to introduce/discuss the related references.

o   …..etc

-       The flow of the paper (especially the first two sections) needs to be reviewed.

-       You need to illustrate whether there are any assumptions made regarding any structural properties of the dataset.

-       Have all the compared models (baseline models) been trained with the same data and experimental settings? It is not clear in the paper (Sections 5.1 and 5.2)

-       You should give the configuration of models of BERT, CNN and BiLSTM.

-       You should explain evaluation metrics based on the problem.

Reviewer 3 Report

The article proposes a dual-channel Chinese Named Entity Recognition approach based on the BERT algorithm.

The approach and its testing are well presented. In the conclusion, the authors may add whether the presented approach is applicable in other areas.

Round 2

Reviewer 1 Report

Usually there is a problem on the dimensions of the LSTM and CNN blocks.

For example, IN the CNN-LSTM case, the outpout of CNN will be [N,32] since 32 filters was used.  What is the ssizeier of each filter?

Reviewer 2 Report

Thank you for addressing all the comments.

Round 3
